# Retinol saturase coordinates liver metabolism by regulating ChREBP activity

Steffi Heidenreich[1], Nicole Witte[1], Pamela Weber[1], Isabel Goehring[1], Alexander Tolkachov[1], Christian von Loeffelholz[2,3,4], Stephanie Döcke[2], Michael Bauer[3,4], Martin Stockmann[5], Andreas F. H. Pfeiffer[2,6], Andreas L. Birkenfeld [7,8], Matthias Pietzke[9], Stefan Kempa[9], Matthias Muenzner[1] & Michael Schupp [1]

The liver integrates multiple metabolic pathways to warrant systemic energy homeostasis. An excessive lipogenic flux due to chronic dietary stimulation contributes to the development of hepatic steatosis, dyslipidemia and hyperglycemia. Here we show that the oxidoreductase retinol saturase (RetSat) is involved in the development of fatty liver. Hepatic RetSat expression correlates with steatosis and serum triglycerides (TGs) in humans. Liver-specific depletion of RetSat in dietary obese mice lowers hepatic and circulating TGs and normalizes hyperglycemia. Mechanistically, RetSat depletion reduces the activity of carbohydrate response element binding protein (ChREBP), a cellular hexose-phosphate sensor and inducer of lipogenesis. Defects upon RetSat depletion are rescued by ectopic expression of ChREBP but not by its putative enzymatic product 13,14-dihydroretinol, suggesting that RetSat affects hepatic glucose sensing independent of retinol conversion. Thus, RetSat is a critical regulator of liver metabolism functioning upstream of ChREBP. Pharmacological inhibition of liver RetSat may represent a therapeutic approach for steatosis.

[1] Charité – Universitätsmedizin Berlin, corporate member of Freie Universität Berlin, Humboldt-Universität zu Berlin, and Berlin Institute of Health, Institute of Pharmacology, Center for Cardiovascular Research, Berlin 10115, Germany. [2] Department of Clinical Nutrition, German Institute of Human Nutrition Potsdam-Rehbruecke, Nuthetal 14558, Germany. [3] Integrated Research and Treatment Center, Center for Sepsis Control and Care (CSCC), Friedrich Schiller University, Jena 07747, Germany. [4] Department of Anesthesiology and Intensive Care, Jena University Hospital, Jena 07747, Germany. [5] Charité – Universitätsmedizin Berlin, corporate member of Freie Universität Berlin, Humboldt-Universität zu Berlin, and Berlin Institute of Health, Department of General, Visceral and Transplantation Surgery, Virchow Campus, Berlin 13353, Germany. [6] Charité – Universitätsmedizin Berlin, corporate member of Freie Universität Berlin, Humboldt-Universität zu Berlin, and Berlin Institute of Health, Department of Endocrinology, Diabetes, and Nutrition, Berlin 10117, Germany. [7] Paul Langerhans Institute Dresden of the Helmholtz Center Munich at University Hospital and Faculty of Medicine, TU Dresden, University Clinic Dresden, Dresden 01307, Germany. [8] Division of Diabetes and Nutritional Sciences, Faculty of Life Sciences and Medicine, King's College London, London SE1 8WA, UK. [9] Integrative Metabolomics and Proteomics, Berlin Institute of Medical Systems Biology/Max- Delbrück Center for Molecular Medicine, Berlin 13125, Germany. Steffi Heidenreich and Nicole Witte contributed equally to this work. Correspondence and requests for materials should be addressed to M.S. (email: michael.schupp@charite.de)

Retinol saturase (RetSat) is an NADH/NADPH- or FADH-dependent oxidoreductase and strongly expressed in liver, adipose tissue and kidney[1]. Major transcriptional regulators of *RetSat* expression include peroxisome proliferator-activated receptor α (PPARα) and forkhead box O1 (FoxO1) in liver[2–4], and PPARγ in adipose tissue, where *RetSat*'s expression is robustly induced during the differentiation of precursor cells into adipocytes[5]. Cellular RetSat protein localizes primarily to the endoplasmic reticulum and catalyzes the conversion of retinol to 13,14-dihydroretinol (13,14-dhretinol)[1, 5, 6], a retinoid metabolite that can act as precursor for the generation of 13,14-dihydro-etinoic acid[7]. Although it was found that these dihydroretinoids bind retinoic acid receptors (RAR) with lower affinities than their non-reduced forms[8], their biological relevance is largely unknown. A recent study identified the 9-*cis*-isomer of 13,14-dihydroretinoic acid as an endogenous ligand for retinoid X receptor (RXR)[9]. Whether RetSat is a biologically relevant enzyme for providing precursors for endogenous RXR ligand generation is currently unknown.

We previously reported that RetSat depletion in adipocyte precursor cells impaired their adipogenic conversion in vitro and found its expression in adipose tissue downregulated in obesity[5]. Surprisingly, reduced adipogenesis upon RetSat depletion was not overcome by providing 1314-dhretinol, suggesting that 13,14-dhretinol generation is not responsible for its pro-adipogenic function. On the other hand, complete inactivation of RetSat's enzymatic activity by mutating the dinucleotide-binding motif blunted its effects, indicating that RetSat promotes adipogenesis via an additional, yet unknown enzymatic reaction[5].

Besides promoting adipogenesis and conferring cellular sensitivity towards oxidative stress[10], deeper insights into RetSat's biological functions are missing. Mice with whole-body germline deletion of RetSat exhibit altered body composition with an unexpected shift to higher fat mass but, so far, have not been associated with any other phenotype[11]. A bioinformatic analysis of gene expression profiles from type 2 diabetes-related animal models and human tissue samples identified *RetSat* as the gene with the highest differential mRNA expression across all analyzed data sets[12], suggesting a much broader involvement of RetSat in regulating glucose and fatty acid metabolism.

To identify novel functions of RetSat we studied the liver, the tissue with the strongest RetSat expression[1, 5]. We found that RetSat expression correlates with liver steatosis in humans and that its hepatic depletion lowers liver triglycerides (TGs) and improves metabolic parameters in dietary obese mice, at least in part, by interfering with the activity of the cellular hexose-phosphate sensor carbohydrate response element-binding protein (ChREBP)[13]. These findings link RetSat to sugar sensing in hepatocytes and may allow for novel therapeutic approaches for metabolic liver diseases.

## Results

**RetSat regulates glycolytic and lipogenic pathways in hepatocytes.** We found RetSat protein robustly expressed in several metabolically relevant organs and highest in liver, followed by kidney, epididymal white adipose tissue and muscle (Fig. 1a, whole blot shown in Supplementary Fig. 9), correlating well with its known mRNA expression pattern[1, 5]. To gain functional insights, we performed gene expression profiling of primary mouse hepatocytes that were depleted of RetSat by siRNA for 48 h (Fig. 1b, whole blot shown in Supplementary Fig. 10). Unexpectedly, RetSat depletion in hepatocytes resulted in a rather high number of 1602 regulated genes (P < 0.05 (two tailed Students's t-test) and fold-change ≥ 1.25 cutoff), suggesting that RetSat's enzymatic function may couple to the control of gene

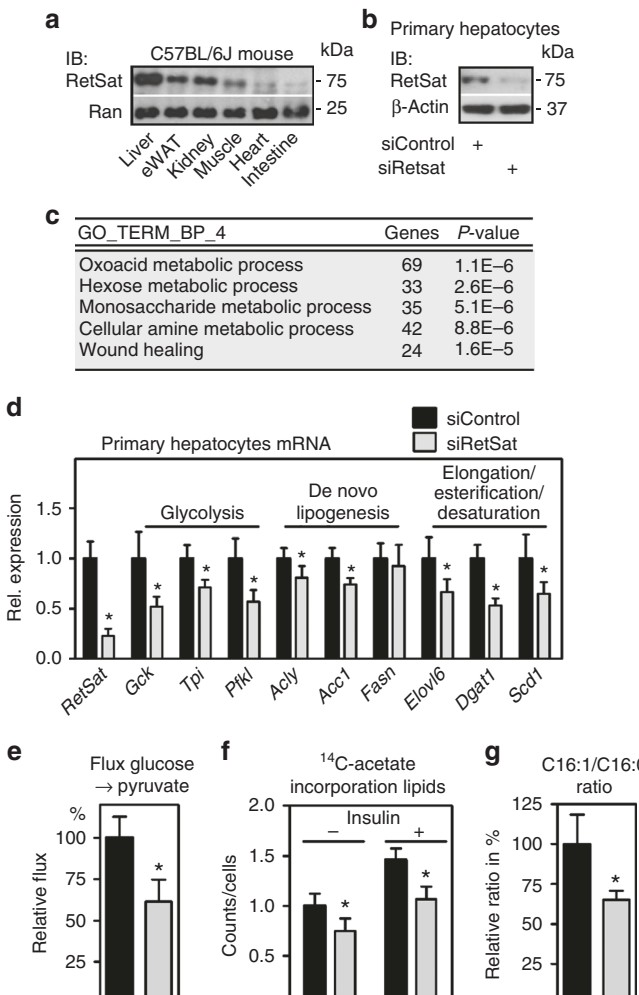

**Fig. 1** RetSat controls glycolytic and lipogenic pathways in primary hepatocytes. **a** RetSat protein expression in metabolically relevant mouse tissues was analyzed by immunoblotting. **b** RetSat protein expression, 48 h after transfecting Control or RetSat siRNA, was determined by immunoblotting. **c** Top five enriched pathways of genes that are regulated by RetSat depletion in hepatocytes were identified by DAVID functional annotation of Affymetrix gene expression profiles. **d** mRNA expression of selected genes in siControl or siRetSat hepatocytes was determined by qPCR. Data are shown as mean ± s.d., n = 6 independent transfections of hepatocyte cultures from two mice; *P < 0.05 by two-tailed t-test. An independent experiments yielded similar results. **e** Hepatocytes depleted of RetSat were incubated with $^{13}$C-glucose for 5 min and $^{13}$C-pyruvate labelling analyzed by mass spectroscopy. Data are shown as mean ± s.d., n = 4 independent transfections of hepatocyte cultures from the same mouse; *P < 0.05 by two-tailed t-test. An independent experiment yielded similar results. **f** Incorporation of $^{14}$C-acetate into extractable lipids was assessed in hepatocytes treated as indicated. Data are shown as mean ± s.d., n = 6 independent transfections of hepatocyte cultures from the same mouse; significance between siControl and siRetSat was tested by two-tailed t-test and *P < 0.05. An independent experiment yielded similar results. **g** Palmitate/palmitoleate ratio in siControl- or siRetSat-treated hepatocytes was assessed by mass spectroscopy. Data are shown as mean ± s.d., n = 4 independent transfections of hepatocyte cultures from the same mouse; *P < 0.05 by two-tailed t-test. An independent experiment yielded similar results

expression. Regulated genes enriched primarily to gene ontology (GO) pathways of intermediary cell metabolism of oxoacids, hexoses/monosaccharides, and amino acids (top five terms shown in Fig. 1c). Among genes related to hexose/monosaccharide metabolism, we noticed that several glycolytic genes, including *aldolase* (*Aldoa*), *phosphofructokinase, liver* (*Pfkl*) and *pyruvate kinase, liver* (*Pklr*) were downregulated upon RetSat depletion (Supplementary Fig. 1a). We validated the regulation of some of these genes by quantitative PCR (qPCR) and extended our analysis to pathways downstream of glycolysis, namely de novo lipogenesis and the elongation, esterification and desaturation of fatty acids. Strikingly, many of these genes were also downregulated (Fig. 1d). We then tested whether altered gene expression bears functional relevance and found a reduction in glycolytic flux, de novo lipogenesis, and a lower palmitoleate/palmitate ratio that indicates reduced palmitate desaturation in RetSat-depleted hepatocytes (Figs. 1e-g). Thus, RetSat depletion impairs glycolytic flux and lipid metabolism in primary hepatocytes.

**RetSat is elevated in obese mouse liver and controls lipid metabolism**. Given this functional relevance, we compared RetSat mRNA levels in livers of lean and diet-induced, obese mice (body weights of $29.7 \pm 1.61$ g vs. $47.9 \pm 2.83$ g, respectively) and found it increased in obese animals (Fig. 2a). We next addressed whether RetSat regulates lipid metabolism in vivo. To acutely deplete RetSat in livers of adult mice, we generated adenoviruses that target βGal (Control) or RetSat by expressing shRNA. Tail-vein injection of adenoviruses yields transient and highly liver-specific expression, as validated by a FLAG-tagged protein that was delivered similarly and detected only in liver (Supplementary Fig. 1b). Depleting hepatic RetSat expression (Fig. 2b, whole blot shown in Supplementary Fig. 11) in adult, NC-fed mice did not change any of the metabolic parameters measured, when analyzed 6 days after virus injection (Supplementary Table 1). In contrast, acute RetSat depletion in mice that were challenged by HS/HFD-feeding before injecting viruses strongly reduced the liver TG content (Fig. 2c, d). Consistently, hepatic expression of genes involved in de novo lipogenesis and fatty acid elongation and desaturation, including the enzyme that produces monounsaturated fatty acids (MUFAs), *stearoyl-CoA desaturase 1* (*Scd1*)[14], was lower (Fig. 2e), resembling the consequences of RetSat depletion in primary hepatocytes. Consistent with reduced expression of genes like *ATP citrate lyase (Acly)*, *acetyl-CoA carboxylase 1 (Acc1)* and *fatty acid synthase (Fasn)* in shRetSat-treated mice, liver de novo palmitate synthesis tended to be lower (Fig. 2f). Food intake of HS/HFD-fed mice with hepatic RetSat depletion was lower without reaching significance (Fig. 2g). Serum levels of TG, non-esterified fatty acids (NEFAs) and blood glucose were strongly reduced in both ad libitum-fed and fasted mice (Fig. 2h-j). Lower blood glucose was not due to increased insulin, since serum insulin was reduced (Fig. 2k). Glucose tolerance was enhanced in mice with liver-specific depletion of RetSat, whereas the glucose-lowering effects of insulin were comparable (Supplementary Fig. 2a, b, respectively). Basal Akt phosphorylation (S473) was slightly higher in shRetSat-treated mice and similar to control mice after the injection of insulin (Supplementary Fig. 2c). Taken together, we found that RetSat expression is elevated in livers of obese mice, and that its acute depleting has profound metabolic effects in HS/HFD-fed, but not in NC-fed mice.

**Hepatic RETSAT correlates with obesity and steatosis in humans**. We then addressed whether RETSAT could be linked to hepatic lipid metabolism in humans. In a set of liver samples derived from abdominal surgery, *RETSAT* mRNA expression correlated positively with patient body mass index (Fig. 3a). Moreover, *RETSAT* expression showed a strong correlation with the degree of steatosis (Fig. 3b), an established clinical parameter determined by histology[15], and the homeostatic model assessment - insulin resistance index (HOMA-IR)[16] (Fig. 3c). We also identified correlations between *RETSAT* expression and serum TG (Fig. 3d), and the percentage of MUFAs from total FA (Fig. 3e). Thus, the pattern of *RETSAT* expression in liver samples could imply a function in hepatic glucose and lipid metabolism in humans.

**RetSat regulates ChREBP and its target gene expression**. We next analyzed livers of HS/HFD-fed mice with or without RetSat depletion in more detail to identify the underlying mechanisms of reduced liver and serum TGs. Increased hepatic de novo lipogenesis contributes to both liver steatosis and elevated serum TG upon HS/HFD-feeding[17, 18]. We therefore tested whether major transcriptional regulators of de novo lipogenesis, sterol regulatory element binding protein 1c (Srebp1c)[19], liver x receptor alpha (Nr1h3 = Lxrα)[20], or ChREBP[21–23] were affected by RetSat depletion. Only *ChREBP* mRNA expression was decreased, with a more pronounced reduction for the weakly expressed, but more active β-isoform[24–26] (Fig. 4a).

By analyzing liver protein expression we found that ChREBPα (migrating at ~100 kDa) was strongly reduced, even more than its mRNA regulation would predict (Fig. 4b, whole blot shown in Supplementary Fig. 12). Protein of the β-isoform (migrating at ~75 kDa)[25] was undetectable under either condition (Fig. 4b). In accordance with lower levels of ChREBP protein, most of the analyzed canonical target genes of ChREBP[27–29] were reduced in livers depleted of RetSat (Fig. 4c). Moreover, the ChREBP target genes *PKLR*[13] and *acetyl-CoA carboxylase 1* (*ACC1*)[30, 31] strongly correlated with *RETSAT* expression in human liver samples (Fig. 4d). Taken together, our data suggest a functional link between RetSat and ChREBP. Thus, RetSat depletion may cause at least some of the observed metabolic alterations by interfering with hepatic ChREBP in HS/HFD-fed mice.

Although NC-fed mice depleted of hepatic RetSat lacked obvious metabolic alterations, we asked whether ChREBP-dependent responses were affected. Maximizing hepatic ChREBP activity by 4 h re-feeding strongly induced the canonical target genes *Fasn*[30, 31] and *regulator of G-protein signaling 16* (Rgs16)[32] in livers of control mice. This induction was significantly impaired in mice with hepatic RetSat depletion (Supplementary Fig. 3a), indicating that hepatic RetSat depletion impairs ChREBP activation also in mice fed NC. Since blood glucose levels during re-feeding were similar in both groups (Supplementary Fig. 3b), reduced ChREBP target gene expression is a primary effect of depleting RetSat and not secondary due to lower blood glucose levels. However, strongly reduced ChREBP protein levels and impaired target gene expression in livers of HS/HFD-fed mice may be due to both, direct effects of RetSat depletion and lower blood glucose levels.

**RetSat controls ChREBP activity and glucose sensing in hepatocytes**. ChREBP is a pivotal regulator of glycolysis and de novo lipogenesis[21]. Our finding that RetSat depletion interferes with ChREBP-dependent pathways is compatible with RetSat's function in primary hepatocytes (Fig. 1). In contrast to liver, RetSat depletion in hepatocytes for 48 h did not affect mRNA or protein levels of ChREBP (Fig. 5a). However, we found an overall downregulation of canonical target genes of ChREBP[27, 28] (Fig. 5b) but not of Lxrα (Supplementary Fig. 4), irrespective of the siRNA target sequences used to deplete RetSat (Fig. 5c). Since ChREBP protein levels were not changed, we asked whether

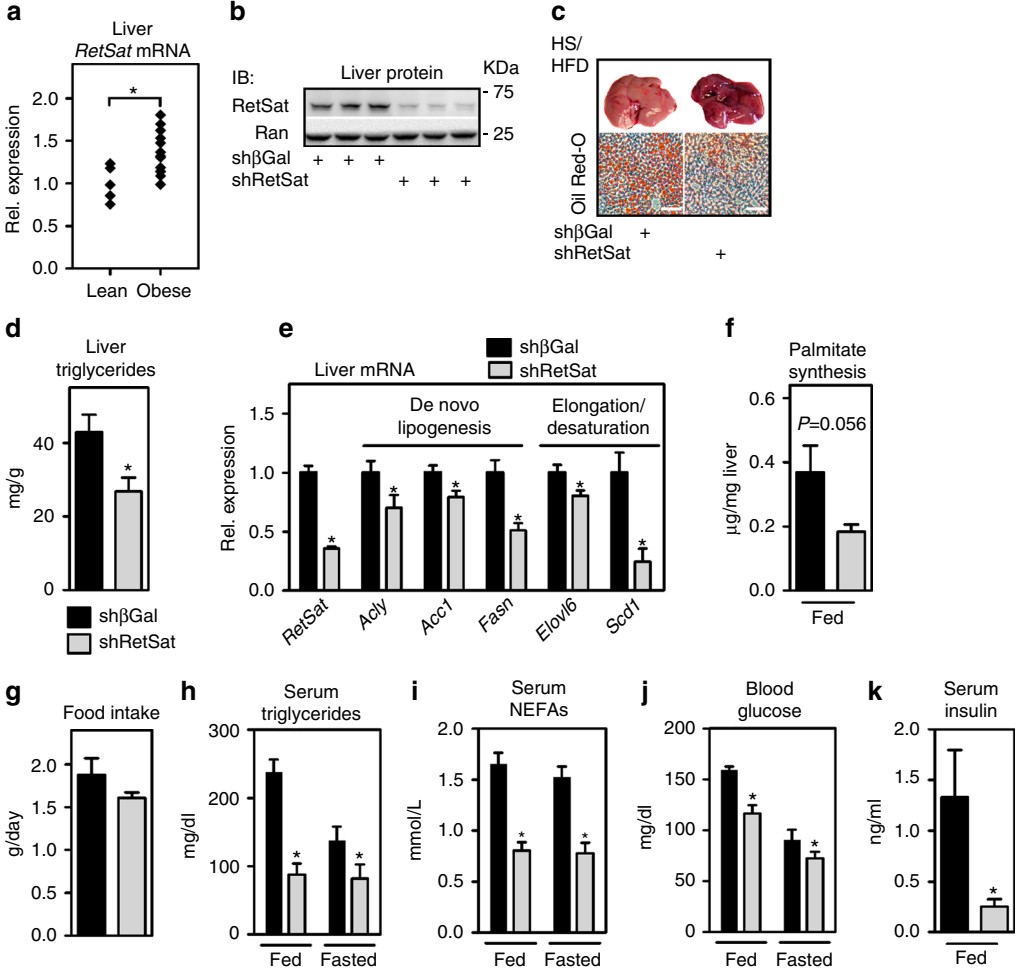

**Fig. 2** Hepatic RetSat expression is increased in obese mice and regulates lipid metabolism. **a** RetSat mRNA expression in livers of lean, NC-fed ($n = 5$) and obese, HS/HFD-fed ($n = 13$) mice was determined by qPCR. *$P < 0.05$ between groups by two-tailed $t$-test. **b** Mice were injected with adenoviruses expressing shRNA targeting βGal or RetSat. Six days later, liver protein was analyzed for RetSat protein by immunoblotting. **c–j** HS/HFD-fed mice were treated as described in **b**, **c** liver stained for TGs by Oil Red-O, scale bars = 200 μM. **d** Liver TGs in mice were determined biochemically. Data are shown as mean ± s.e.m., $n = 12$ (shβGal), 13 (shRetSat); *$P < 0.05$ by two-tailed $t$-test. An independent experiment yielded similar results. **e** Hepatic mRNA expression of genes involved in lipid metabolism was determined by qPCR. Data are shown as mean ± s.e.m., $n = 7$ (shβGal), 6 (shRetSat); *$P < 0.05$ by two-tailed $t$-test. An independent experiment yielded similar results. **f** Newly synthesized palmitate in liver was assessed by determining the incorporation of deuterated water. Data are shown as mean ± s.e.m., $n = 5$ (shβGal), 5 (shRetSat); $P$ value was determined by two-tailed $t$-test. **g** Food intake was measured for two 24 h periods. Data are shown as mean ± s.e.m., $n = 6$ (shβGal), 5 (shRetSat); *$P < 0.05$ by two-tailed $t$ test. **h–j**: serum TGs, NEFAs, and blood glucose in ad libitum-fed or 24 h-fasted mice, determined 6 days after virus injection. Data are shown as mean ± s.e.m., $n = 13$ (shβGal), 12 (shRetSat); *$P < 0.05$ between both groups by two-tailed $t$-test. An independent experiment yielded similar results. **k** Serum insulin in ad libitum-fed mice was determined by ELISA. Data are shown as mean ± s.e.m., $n = 9$ (shβGal), 10 (shRetSat); *$P < 0.05$ by two-tailed $t$-test

RetSat depletion reduces ChREBP activity. Indeed, a carbohydrate response element (ChoRE)-driven luciferase reporter[33], showing the expected induction by high glucose, was less active in RetSat-depleted hepatocytes (Fig. 5d). Moreover, ChREBP target genes matched this activity pattern in primary hepatocytes when exposed to either low or high glucose concentrations (Fig. 5e).

**RetSat depletion impairs the nuclear accumulation of ChREBP.** Upon exposure to high glucose concentrations, cyto-solic ChREBP translocates and accumulates in the nucleus[34]. After validating the ChREBP antibody for immunofluorescence (adenoviral ChREBP over-expression strongly induced ChREBP staining, Supplementary Fig. 5a), we compared the nuclear signal intensities of endogenous ChREBP in primary hepatocytes exposed to low or high glucose concentrations. Insulin was added to maximize ChREBP nuclear translocation[35]. We found that high glucose conditions caused a moderate increase in nuclear

ChREBP staining (Fig. 6a). In hepatocytes depleted of RetSat (Supplementary Fig. 5b), nuclear accumulation of ChREBP upon high glucose was blunted (Fig. 6a, arrows, and quantification in Fig. 6b). We then asked whether the putative enzymatic product 13,14-dhretinol can compensate for impaired ChREBP responses due to RetSat depletion. Surprisingly, neither the reduced induction of ChREBP target genes (Fig. 6c) nor lower de novo lipogenesis (Fig. 6d) were rescued by providing 13,14-dhretinol, suggesting that dhretinol formation is not the enzymatic reaction required for ChREBP's activity. We next asked whether a lack of glucose metabolites could be responsible for reduced ChREBP activity and found that RetSat-depleted cells contained higher levels of certain glycolytic and pentose phosphate pathway metabolites, including glucose 6-phosphate and sedoheptulose 7-phosphate (Supplementary Fig. 6). This argues against a primary defect in the generation of ChREBP-activating glucose metabo-lites in RetSat-depleted hepatocytes. Interestingly, increased

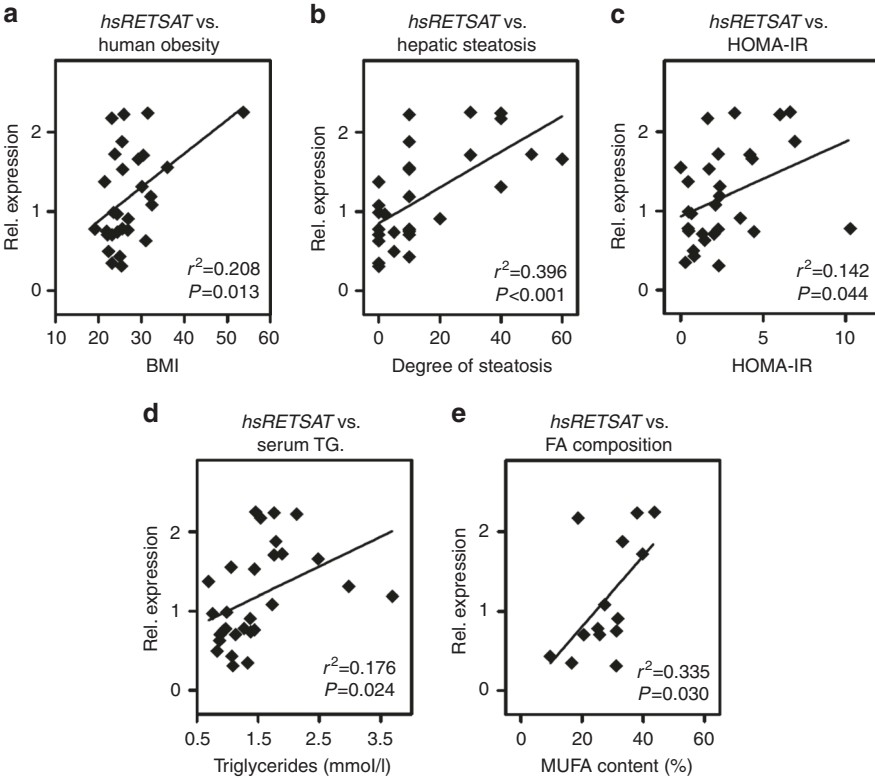

**Fig. 3** Hepatic RetSat expression in humans correlates with obesity and liver steatosis. *RETSAT* mRNA expression in human liver samples was determined by qPCR ($n = 29$) and correlated with **a** patient body mass index, **b** degree of hepatic steatosis, **c** HOMA-IR and **d** serum TG. **e** In a subset of 14 liver samples, MUFA content was determined and correlated with the expression of *RETSAT*. Normality of data was assessed by the Kolmogorov–Smirnov test and, depending on data distribution, significance was determined by Pearson **a**, **c**–**e** or Spearman **b** correlation coefficient

concentrations of glucose 6-phosphate were detected also in ChREBP-depleted[36] or ChREBP-deleted livers[37], most likely due to altered/impaired glucose-6-phosphate turnover. On the other hand, adenoviral over-expression of ChREBP led to a partial recovery of ChREBP target genes that were downregulated by RetSat depletion (Fig. 6e, f). Notably, some of the genes downregulated by RetSat depletion, including *glucokinase* (*Gck*), were not rescued by ectopic ChREBP (*Gck* mRNA reduced to 51.73 ± 15.79% in siRetSat versus siControl, and similarly reduced to 45.12 ± 20.51% in siRetSat + Adeno-ChREBP versus siControl hepatocytes), suggesting that there are additional mechanisms by which RetSat depletion interferes with gene expression.

We finally compared BioGPS profiles[38, 39] of RetSat and ChREBP and found a strikingly similar mRNA expression pattern in different murine organs and cell types (Supplementary Fig. 7). To evaluate whether RetSat links functionally to ChREBP in other cell types, we depleted RetSat in mature 3T3-L1[40] adipocytes that, like hepatocytes, robustly express both proteins (Supplementary Fig. 8a). Indeed, the high-glucose induced upregulation of the ChREBP target gene *Acly* was strongly impaired in RetSat-depleted adipocytes (Supplementary Fig. 8b), suggesting that RetSat's control of ChREBP activity is not restricted to hepatocytes.

## Discussion
We discovered a novel function of RetSat in liver metabolism. RetSat depletion in primary hepatocytes reduced glycolytic flux, de novo lipogenesis and palmitate desaturation. In HS/HFD-fed mice, acute depletion of RetSat reduced liver TG content, improved glucose tolerance, and lowered blood glucose and serum lipids. Moreover, we found that RetSat expression in liver correlates positively with body weights in both mice and humans,

consistent with increased hepatic de novo lipogenesis in obese, insulin-resistant subjects[17].

To identify the mechanism by which RetSat has such a profound impact on metabolic pathways, we profiled gene expression in primary hepatocytes. At first we were surprised to find such a high number of genes regulated by the sole depletion of this enzyme. However, this pattern hinted towards a transcriptional regulator that would function downstream of RetSat. We identified this transcriptional regulator as ChREBP. We found that RetSat is required for (1) ChREBP activity, target gene expression, and its nuclear accumulation upon high glucose exposure in primary hepatocytes, (2) ChREBP target gene expression in livers of mice re-fed with NC and (3) ChREBP protein levels and target gene expression in livers of HS/HFD-fed mice. Moreover, RetSat expression correlates strongly with that of canonical ChREBP target genes in human liver samples. Most importantly, acute depletion of RetSat in livers of HS/HFD-fed mice induced a phenotype that, in many respects, carries strong resemblance with that of acute depletion of ChREBP in genetically obese mice, including a reduction in serum NEFAs and lower blood glucose levels[36]. On the basis of these findings, our current working model places RetSat upstream of ChREBP by catalyzing the generation of a metabolite that enhances ChREBP's nuclear translocation, or by degrading a molecule that mediates cytosolic retention.

RetSat's regulation of ChREBP activity may be relevant in other cell types since there is a strong overlap in their tissue expression pattern. Indeed, we show that RetSat depletion interferes with glucose sensing and the expression of ChREBP target genes also in adipocytes. Since we recently reported that activating ChREBP by high glucose concentrations or expressing a constitutive-active ChREBP in precursor cells enhances adipocyte differentiation[25], RetSat's requirement for in vitro adipocyte differentiation[5]

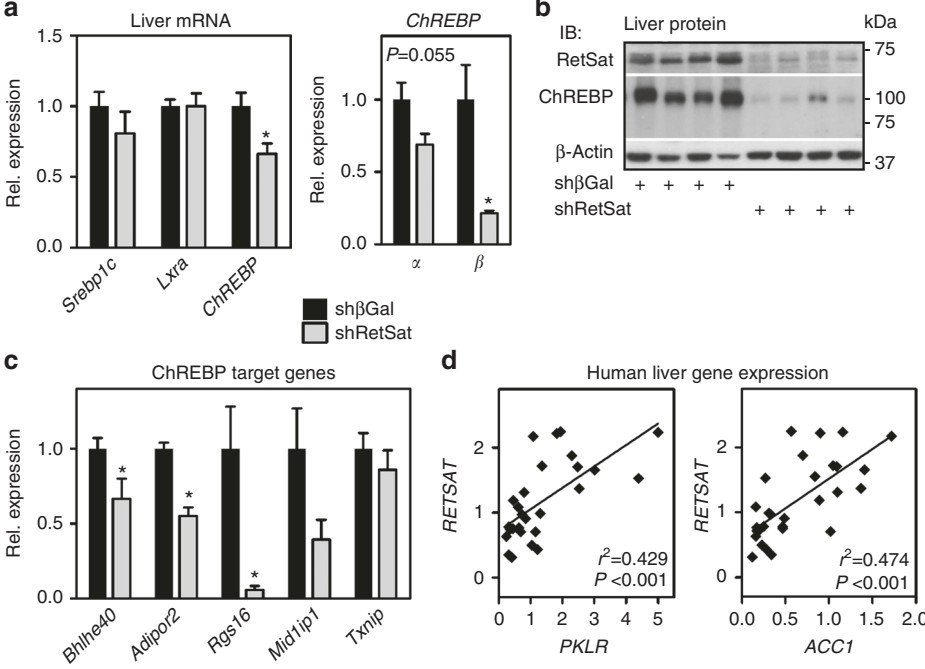

**Fig. 4** RetSat depletion in mouse liver reduces protein levels and target gene expression of ChREBP. Mice fed HS/HFD were injected with adenoviruses expressing shRNA targeting βGal or RetSat. Six days later, livers were analyzed for **a** mRNA expression of the indicated transcription factors. **b** RetSat and ChREBP protein expression by immunoblotting, and **c** known ChREBP target genes by qPCR. In **a**, **c**, data are shown as mean ± s.e.m., $n = 7$ (shβGal), 6 (shRetSat); $*P < 0.05$ by two-tailed $t$-test. **d** RETSAT, PKLR and ACC1 mRNA expression in human liver samples was determined by qPCR ($n = 29$) and correlated. Normality of data were tested by the Kolmogorov–Smirnov test and significance was determined by the Pearson correlation coefficient

may in fact be due to its ChREBP-activating function. Reduced RetSat expression in adipose tissue of obese mice and humans[5] is also consistent with its function in de novo lipogenesis, since obesity and insulin resistance, in stark contrast to liver, associate with reduced adipose tissue de novo lipogenesis[17, 24].

Although Srebp1c and Lxrα are interconnected with ChREBP and partially overlap in controlling glycolytic and lipogenic gene expression[31, 41–43], we have no indication that RetSat directly affects either of them. However, we would expect that Srebp1c's proteolytic activation, functionally more relevant than its mRNA levels, could be affected in HS/HFD-fed mice depleted of hepatic RetSat due to low serum insulin levels which are known to affect its processing[44].

By identifying RetSat's function in liver several questions arise that should be addressed in future experiments. The most important is in regard to RetSat's enzymatic function. Similar to adipocyte differentiation[5], defects due to RetSat depletion were not overcome by providing the putative enzymatic product 13,14-dihretinol. If not retinol conversion, by which reaction does RetSat control ChREBP activity? Molecular cues, including post-translational modifications, that enhance or prevent nuclear translocation of ChREBP are complex[27, 45–47] and a careful dissection may identify RetSat's involvement. For instance, it has been hypothesized that RetSat reduces $\Delta^{13}$ of certain polyunsaturated fatty acids (PUFAs)[48]. Since ChREBP activity is suppressed by PUFAs, but not by MUFAS or saturated fatty acids[35], RetSat may induce ChREBP activity by metabolizing PUFAs. Notably, not all effects of RetSat depletion can be explained by working upstream of ChREBP, indicating that additional mechanisms may play a role. Another puzzling aspect is that RetSat whole-body knockout mice, even when fed a HFD, lacked changes in hepatic or serum TG levels[11]. Whether or not this is due to compensatory mechanisms that can be invoked in germline knockout models has to be determined.

Taken together, we found that RetSat coordinates liver metabolism by upstream regulation of ChREBP. When balanced with metabolically favorable aspects of activated ChREBP, e.g. de novo lipogenesis in adipose tissue[24], and fatty acid desaturation[49] and suppression of SREBP2-mediated detrimental cholesterol overload[50] in liver, pharmacological inhibition of RetSat and lower ChREBP activity in liver may have therapeutic potential to treat hepatic steatosis, dyslipidemia and hyperglycemia.

## Methods

**Human liver samples**. Informed consent was obtained from each patient and the study protocol conformed to the ethical guidelines of the 1975 Declaration of Helsinki and was approved by the ethics committee of the Charité -Universitätsmedizin Berlin (INSIGHT study, German Clinical Trials Register: DRKS00005450). Twenty-nine patients were eligible for the current study, details regarding inclusion and exclusion criteria have been published[51]. To avoid hypoxia-induced artefacts, perfused liver tissue samples were taken by knife extraction immediately after starting surgery. Degree of liver steatosis was evaluated by means of histopathology according to standard criteria[15]. Liver samples were immediately snap-frozen and processed for mRNA and gene expression. Characteristics of patients are given in Supplementary Table 2. In a subset of higher yield liver samples ($n = 14$), lipid profiles were determined based on a published methodology[52]. Lipid extracts were methylated by acid- and base-catalyzed procedures using a combination of 0.5 N methanolic sodium hydroxide (Merck) and 10% (w/w) boron trifluoride-methanol (Supelco, USA, 100 °C for 5 min each). Subsequently, fatty acid methyl esters (FAME) were purified by thin layer chromatography and dissolved in n-hexane for analysis. A system of two gas chromatography/flame ionization detector methods was used to analyze the full FA spectrum (GC-17 V3 Shimadzu, DB-225MS Agilent and GC-2010, Shimadzu, CP-select, Varion). FA data are presented as percentage of the total area of all FA peaks (% of total FAME).

**Animal studies**. Animal procedures were in accordance with institutional guidelines and approved by the corresponding authorities (approval at the University of Pennsylvania by the Institutional Animal Care and Use Committee, and at the Charité by the Landesamt für Gesundheit und Soziales Berlin). Male C57BL/6 J mice (provided by the Jackson Laboratory or the Research Institute for Experimental Medicine of the Charité) aged 6–8 weeks were used for all experiments if not stated otherwise. Mice were fed either normal chow (NC, ssniff R/M-H) or a 60 kcal% high-fat/high-sucrose diet (HS/HFD, D12492, Research Diets). Obese and

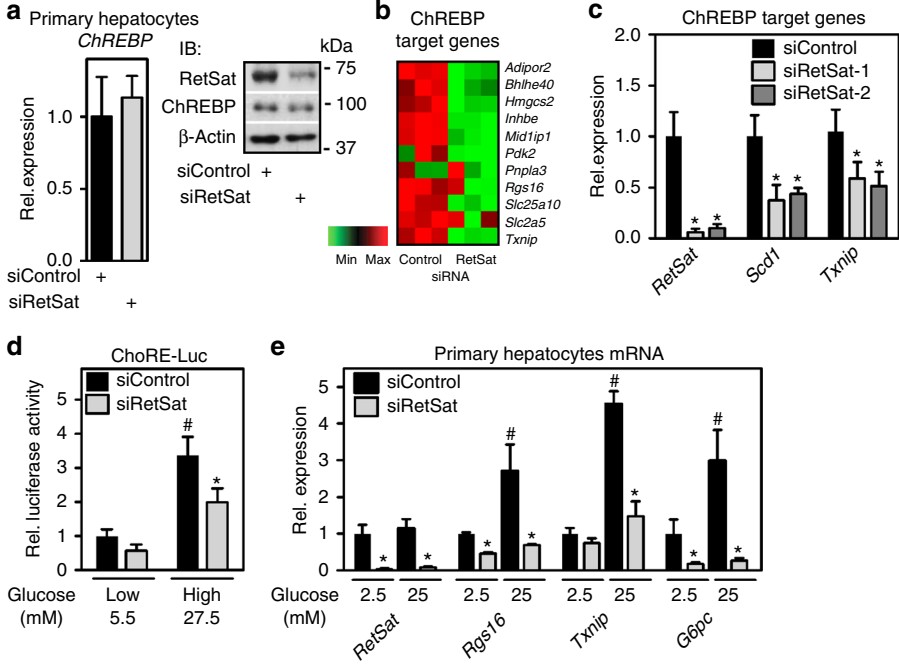

**Fig. 5** RetSat controls ChREBP activity and glucose sensing in primary hepatocytes. Primary mouse hepatocytes were treated with Control or RetSat siRNA for 48 h, (**a**, *left*) *ChREBP* mRNA expression determined by qPCR, and RetSat and ChREBP protein levels determined by immunoblotting (**a**, *right*). *left*, Data are shown as mean ± 11s.d., $n = 3$ independent transfections of hepatocyte cultures from the same mouse. Two independent experiments yielded similar results. **b** Hepatocytes were treated as described in **a** and mRNA expression of a selection of known ChREBP target genes visualized in a heatmap. **c** Primary hepatocytes were depleted of RetSat using two siRNA's targeting different sites of the RetSat transcript for 48 h, and expression of the indicated genes analyzed by qPCR. Data are shown as mean ± s.d., $n = 6$ independent transfections of hepatocyte cultures from two different mice; *$P < 0.05$ between siControl und siRetSat by one-way ANOVA with Bonferroni post test. An independent experiment yielded similar results. **d** Hepatocytes treated with Control or RetSat siRNA were transfected with a ChoRE-Luc reporter, exposed to low and high glucose concentrations as indicated, and analyzed for luciferase activity. **e** Hepatocytes treated with Control or RetSat siRNA were exposed to low and high glucose concentrations as indicated, and mRNA expression determined by qPCR. In **d**, **e**, data are shown as mean ± s.d., $n = 6$ independent transfections of hepatocyte cultures from two mice; two-way ANOVA with Bonferroni post test revealed significances between low and high glucose concentrations (#$P < 0.05$) and between siControl and siRetSat (*$P < 0.05$). An independent experiment yielded similar results

insulin-resistant mice were obtained feeding 6-week-old mice HS/HFD for at least 8 weeks. For fasting experiments, cages were replaced and food withheld for 24 h. Re-feeding experiments were carried out by providing chow diet for 4 h after the 24 h fasting period. For comparing hepatic RetSat expression in lean or diet-induced obese mice, a previously described cohort was used[5].

**Adenovirus generation and tail-vein injection into mice**. Adenoviral vectors expressing shRNA against β-Galactosidase (βGal) or RetSat were cloned using the BLOCK-iT U6 RNAi Entry Vector Kit and Adenoviral RNAi Expression System according to the manufacturer's instructions (ThermoFisher). Adenoviruses expressing GFP or a GFP- murine ChREBP fusion protein are described elsewhere[25]. Viruses were amplified in HEK293 cells (Clontech) and purified by standard CsCl gradient centrifugation and dialyzed against 0.9% saline. Titers were determined by the Adeno-X Rapid Titer Kit (Clontech). For liver-selective RetSat depletion, ~10E10 infectious units of shβGal-or shRetSat-expressing adenoviruses were injected via the tail vein into lean or obese, insulin-resistant mice.

**Mouse characterization**. 24 h food intake was determined three days after virus injection. Five days after injection, mice were fed ad libitum or starved for 24 h, blood glucose determined (Contour, Bayer) and sacrificed. Serum was obtained by cardiac puncture and organs collected for protein and mRNA analyses. Liver TG were determined after hydrolyzing ~50 mg of liver tissue for 5 h at 60 °C with 1:1 volume ethanol/30% KOH. After adding equal volumes of 1 M $MgCl_2$ and 10 min incubation on ice, samples were centrifuged and TG-derived glycerol content in the liquid phase determined (Diasys). Serum TG (Diasys), NEFAs (Wako diagnostics) and insulin (Rat Insulin ELISA, CrystalChem) were measured by the indicated kits. For histology, liver pieces were fixed overnight in 4% paraformaldehyde and embedded in OCT for cryosectioning and subsequent Oil Red-O staining. Glucose tolerance was determined by intraperitoneal (ip.) injection of 0.5 g/kg glucose after a 16 h fasting period and repeated blood glucose measurements. Insulin tolerance was assessed by ip. injection of 0.75 U/kg insulin (Insuman rapid, Sanofi) after a 6 h fasting period and blood glucose measurements after the indicated times.

**Measurement of hepatic de novo lipogenesis**. Mice were fed ad libitum and ip. injected with deuterated water (Sigma) (20 μl per gram body weight). Mice were killed 6 h later and liver tissue and whole blood collected. Palmitate content was analyzed using gas chromatography-mass spectrometry. We determined the percentage contribution of newly made fatty acid using the equation: percentage of newly made fatty acid = (total $^2$H-labeled fatty acid/($^2$H-labeled body water × $n$)) × 100, where $n$ is the number of exchangeable hydrogens, assumed to equal 22 for palmitate. We determined the absolute amount of newly synthesized fatty acids by multiplying the percentage of newly made fatty acids by the concentration of the total fatty acids[53].

**Primary hepatocyte isolation and cell culture**. Isolation of hepatocytes was performed as previously described[54] with slight modifications. Mice were anesthetized by ip. injection of ketamine-xylazine and livers perfused via the inferior vena cava with 25 ml of pre-warmed perfusion buffer (Earle's Balanced Salt Solution w/o $CaCl_2/MgCl_2$ (ThermoFisher)) followed by perfusion with 30 ml of pre-warmed digestion buffer (Hank's Salt Solution w/o Phenol red (ThermoFisher), supplemented with 5000 U collagenase (Worthington)). The excised liver was minced, the cell suspension filtered through a 250 micron mesh filter, and hepatocytes collected by a Percoll gradient centrifugation (GE Healthcare). Cell viability was assessed by Trypan Blue staining and hepatocytes seeded on collagen-coated 12-well plates (250,000 cells per well) in Dulbecco's modified Eagle's medium (DMEM) w/o pyruvate containing 25 mM glucose, 10% fetal bovine serum (FBS) (Invitrogen) and 1% penicillin/streptomycin (GIBCO). 3T3-L1 cells (ATCC) were grown to confluence in DMEM w/o pyruvate containing 25 mM glucose, 10% FBS, and penicillin/streptomycin (ThermoFisher). Adipocyte conversion of 3T3-L1 cells was induced by supplementing 10 μg/ml insulin, 2 μM dexamethasone and 500 μM isobutylmethylxanthine for 2 days and for another 2 days by 10 μg/ml insulin only[55]. Cells were used for experiments when at least 90% were differentiated to adipocytes. HEK293 cells were grown in DMEM containing 10% FBS and 1% penicillin/streptomycin. Glucose concentrations for sensing experiments were used as indicated. Cells were not tested for mycoplasma contamination.

**siRNA-mediated silencing in hepatocytes and 3T3-L1 adipocytes**. After the attachment of hepatocytes to the cell culture plate, media was replaced by 500 μl of

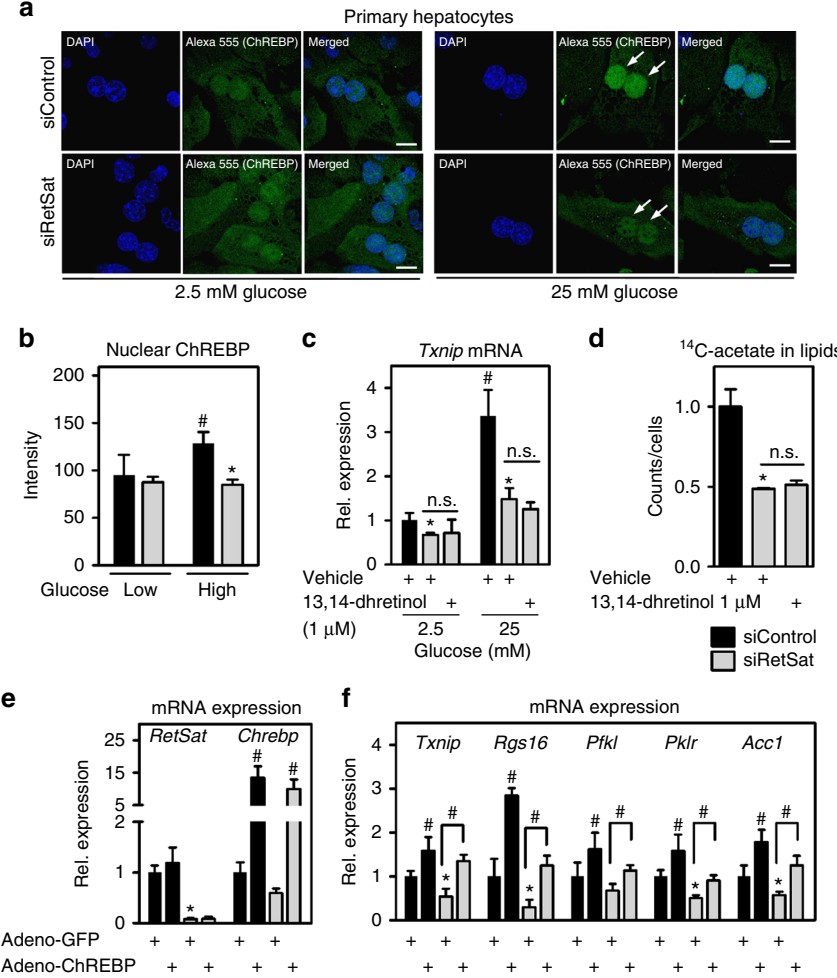

**Fig. 6** RetSat depletion prevents the glucose-induced nuclear accumulation of ChREBP independent of 13,14-dihydroretinol generation. **a** Primary hepatocytes were seeded on cover slips in 2.5 mM glucose. 24 h later, hepatocytes were treated with Control or RetSat siRNA overnight. The next day hepatocytes were exposed to 2.5 or 25 mM glucose and insulin as indicated for 24 h. After fixation, endogenous ChREBP was stained by immunocytochemistry and its localization determined by confocal microscopy. Nuclei were stained using DAPI, *scale bars* = 20 μm. Representative images of three independent experiments are shown. **b** Quantification of nuclear staining. Data are shown as mean ± s.d., *n* = 6 random optical fields of averaged nuclei intensities (total of >15 nuclei for each condition) from hepatocyte cultures from the same mouse; two-way ANOVA with Bonferroni post test revealed significances between low and high glucose (#$P < 0.05$) and between siControl and siRetSat (*$P < 0.05$). **c** Hepatocytes were treated with Control or RetSat siRNA overnight. The next morning, cells were incubated with vehicle (DMSO) or 1 μM 13,14-dhretinol for 24 h at the indicated glucose concentrations and mRNA expression of *Txnip* determined by qPCR. Data are shown as mean ± s.d., *n* = 4. Two-way ANOVA with Bonferroni post test revealed significances between low and high glucose (#$P < 0.05$) and between siControl and siRetSat (*$P < 0.05$), treatment with 13,14-dhretinol had no effect. **d** Incorporation of [14]C-acetate into extractable lipids was assessed in hepatocytes depleted of RetSat for 48 h and supplemented with 13,14-dhretinol for the final 24 h. Data are shown as mean ± s.d., *n* = 4 independent transfections of hepatocyte cultures from the same mouse; *$P < 0.05$ between siControl und siRetSat by one-way ANOVA with Bonferroni post test. **e**, **f** Primary hepatocytes were treated with Control or RetSat siRNA and adenoviruses expressing GFP or a GFP-ChREBP fusion protein. Forty-eight hours after transfection/infection, **e** mRNA expression of *RetSat* and *ChREBP* and **f** ChREBP target genes were analyzed by qPCR. Data are shown as mean ± s.d., *n* = 6 independent transfections/infections of hepatocyte cultures from two mice; two-way ANOVA with Bonferroni post test showed significances between GFP and ChREBP (#$P < 0.05$) and between siControl and siRetSat (*$P < 0.05$). n.s., not significant

DMEM without supplements, and cells transfected with 1 nmol of siRNA (Eurogentec) and 4 μl of Lipofectamine 2000 (Invitrogen) per 12-well overnight. The next morning, media was replaced by DMEM containing 10% FBS and 1% penicillin/streptomycin. siRNA-mediated silencing in 3T3-L1 adipocytes was performed by electroporation (Nucleofector solution V, Lonza) with 3 nmol of siRNA (Eurogentec, Supplementary Table 3).

**GC-MS based metabolomics and de novo lipogenesis in hepatocytes**. Hepatocytes were harvested using cooled (−20 °C) 50% MeOH/H₂0. To analyze incorporation of [13]C carbon in central glycolytic intermediates, cells were incubated for 5 min with [13]C6 glucose (Sigma) and label incorporation was analyzed as previously described[56]. Extracts were transferred to falcon tubes and chloroform was added to achieve mixture of methanol-chloroform-water (MCW) (5:2:1/v:v:v)

(Methanol LC-MS-grade, Chloroform Reagent Plus 99,8% Sigma-Aldrich); cinnamic acid as internal standard (Sigma-Aldrich) was added (2 mg/ml). Supernatant was collected and shaken at 750 r.p.m. and 4 °C for 60 min followed by centrifugation for 10 min at 5000 *g* to separate the polar (top), lipid (bottom) and interface (tissue debris) layers. Polar and lipid phases were dried under vacuum for 12 h. Polar and lipid extracts were processed and analyzed using gas chromatography-mass spectrometry (GC-MS)[57]. Metabolite analyses were performed by a Pegasus III mass-spectrometer (LECO, St. Joseph, USA) equipped with an Agilent 6890 N gas chromatograph and a VF-5ms column with 30 m length and 250 μm inner diameter (Agilent, Santa Clara, USA). 1 μl of sample was injected into a baffled liner (Gerstel, München, Germany) with a 1:5 split-ratio under a helium-flow of 1.2 ml/min. The oven was heated from 70 to 350 °C with 5 °C/min to 120 °C and 7 °C/min to 350 °C followed by 2 min hold time. Scan rates of 20 Hz and mass ranges of 70–600 Da were used. The GC-MS chromatograms were initially

processed with the ChromaTOF software (LECO). The Golm metabolome database (GMD) was used to identify substances in chromatograms achieved from polar and lipid phases with respect to spectra-similarity and retention index. Data matrices for relative quantification were extracted from the mass spectra using MetMax software[57] and metabolite data were normalized to the internal standard measured in the corresponding sample. Lipogenesis was determined by incubating primary hepatocytes with 0.25 μCi/ml [1-$^{14}$C]-acetic acid and 25 mM glucose with or without 100 nM insulin for 2 h. Cells were then washed 3× with PBS and harvested in 1:1 volume ethanol/30% KOH. After saponification at 70 °C for 2 h and acidification with 5 M sulfuric acid, total lipids were extracted by hexane and incorporated $^{14}$C radioactivity measured by scintillation.

**Confocal microscopy**. Cells were fixed with 4% paraformaldehyde on ice for 20 min, washed with PBS and permeabilized with 0.5% Triton X-100 at room temperature (RT) for 15 min. To block non-specific staining, cells were washed and incubated for 1 h at RT in PBS containing 3% BSA and 0.02% Tween and immunolabeled overnight at 4 °C with primary antibodies: RetSat[5] (diluted 1:300 in 3% BSA with 0.02% Tween20), ChREBP (NB400-135, Novus Biologicals, lot M1, diluted 1:300 in 3% BSA with 0.02% Tween20). Cells were washed and incubated with fluorochrome-conjugated secondary antibody (Alexa Fluor® 555 (Life Technologies, diluted 1:600 in 3% BSA) for 1 h at RT. Cells were washed and nuclei were stained with DAPI (Life Technologies) and mounted in Mowiol. Cells were imaged using a confocal microscope (Leica DM 2500). Background subtraction was performed and intensities were quantified by ImageJ software.

**Luciferase reporter assays**. Following transfection of primary hepatocytes with control or RetSat siRNA overnight, cells were transfected with a reporter that contained two copies of the ACC1 carbohydrate response element (ChoRE) in its promoter driving firefly luciferase expression[33] using lipofectamine in the presence of 5.5 mM glucose and 100 nM insulin for 4 h. Culture medium was changed to either 5.5 or 27.5 mM glucose and 100 nM insulin and cells harvested 16 h later. ChoRE-firefly luciferase reporter activity was normalized to co-expressed renilla luciferase (Dual Luciferase, Promega).

**Immunoblotting**. Whole-tissue or whole-cell proteins were isolated and homogenized by standard methods and separated by sodium dodecyl suphate–polyacrylamide gel electrophoresis. Protein concentrations were determined by the BCA method (Thermo Scientific). After incubation with antibodies for RetSat[5] (diluted 1:1000 in 4% skimmed milk), ChREBP (NB400-135, lot J1,M1 or M6 (Novus Biologicals, diluted 1:1000 in 0.05% BSA)[25] (both antibodies were validated in the referenced studies), Ran (#610340, BD Biosciences, diluted 1:1000 in 4% skimmed milk), Anti-flag (A8592, Sigma, diluted 1:1000 in 4% skimmed milk), β-Actin (sc-47778, Santa Cruz, diluted 1:1000 in 4% skimmed milk), phospho Akt (S473, #4060, Cell Signaling, diluted 1:1000 in 4% BSA) and total Akt (#9272, Cell Signaling, diluted 1:1000 in 4% BSA), secondary horseradish-conjugated antibodies (goat anti-rabbit (#31460, Pierce) and goat anti-mouse (#31440, Pierce), diluted 1:1000-1:5000 in 4% skimmed milk) were added as appropriate and a chemiluminescent substrate kit (Thermo) used for detection.

**Isolation and quantification of mRNA expression**. RNA was purified using spin column kits (Macherey-Nagel, Germany). cDNA was generated using MMLV-RT (Promega) or the High Capacity cDNA Reverse Transcription Kit (Applied Biosystems). qPCRs were performed using Sybrgreen PCR Mastermix (Roche) and evaluated according to the standard curve method. All mRNA expression data were normalized to murine 36B4 or human HPRT. Primer sequences are listed in Supplementary Table 3. Gene expression profiling was performed using Affymetrix Mouse Gene 1.0 ST Array arrays by hybridizing three independent transfections of control or RetSat siRNA-treated primary hepatocytes as described previously[5] by the University of Pennsylvania Microarray Core Facility. Gene Ontology analysis of regulated genes was performed using DAVID[58, 59]. Row-normalized heatmaps were generated by the Heatmap Builder.

**Statistical analyses**. Statistical tests are described in the respective figure legends. Sample size estimate for mouse studies was based on power analyses using the analysis platform at the University of Muenster (https://campus.uni-muenster.de/fileadmin/einrichtung/imib/lehre/skripte/biomathe/bio/falla.html). With the observed group differences and the expected variation of measured in vivo parameters, $n = 5$ will provide more than 90% power at the type I error rate of 0.05. Mouse experiments were not randomized or conducted blind to conditions, no exclusion criteria were defined. After testing for normal distribution, statistical analysis was performed using one-way analysis of variance (ANOVA) followed by Bonferroni post test when more than two groups were compared, two-way ANOVA followed by Bonferroni post test when two conditions were investigated and a two tailed Students's $t$ test when only two groups of data were concerned, and $P < 0.05$ was deemed significant. Correlation analyses regarding human data were performed using the Pearson or Spearman rank correlation coefficient. Statistical analyses were calculated by GraphPad Prism (GraphPad Software).

**Data availability**. Affymetric expression profiling data of primary hepatocytes depleted of RetSat that support the findings of this study have been deposited in the Gene Expression Omnibus (GEO) repository[60] with the accession code GSE100211.

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

## Acknowledgements

This work was supported by the German Research foundation (DFG, Emmy Noether grant SCHU 2546/1-1) and a Career Integration Grant from the European Union (CIG 291867) to M.S. as well as by grants from the DFG BI1292/4-2 and DFG IRTG2251 to A.L.B. This study was initiated in the laboratory of Mitch Lazar (Perelman School of Medicine, University of Pennsylvania, Philadelphia) and we are deeply grateful for his support. We thank J. Millar and the Metabolic Tracer Resource at the Penn Institute for Diabetes, Obesity and Metabolism for the lipogenesis assay in mouse liver. We thank Howard Towle (University of Minnesota, Minneapolis) for plasmids and helpful discussions.

## Author contributions

S.H., N.W., P.W., I.G., A.T., C.v.L., S.D., M.P., M.M. and M.S. researched data. S.H., N.W., C.v.L., S.D., S.K., M.B., M.S., A.F.H.P., and A.L.B and M.S. processed, discussed or evaluated the data and edited the manuscript. S.H. and M.S. wrote the manuscript.

## Additional information

**Competing interests:** The authors declare no competing financial interests.

