## [Peer Review File · Nature Communications]

Reviewers' comments:

Reviewer #1 (Remarks to the Author):

In this study, the authors implicate the oxidoreductase retinol saturase (RetSat) in metabolic liver diseases. Hepatic RetSat expression correlated with steatosis and serum triglycerides in humans. Liver-specific disruption of RetSat in dietary-obese mice lowered hepatic and circulating triglycerides and normalized hyperglycemia. Mechanistically, RetSat depletion reduced the activity of carbohydrate response element-binding protein (ChREBP), a cellular hexose-phosphate sensor and inducer of de novo lipogenesis. Defects upon RetSat depletion were rescued by ectopic ChREBP but not by its putative enzymatic product 13,14-dihydroretinol, suggesting that RetSat affects hepatic glucose sensing independent of retinol conversion.

Altogether, the current study suggests that RetSat acts as a critical regulator of liver metabolism which functions upstream of ChREBP.

Major comments

1-Metabolic and physiologic analysis of mice with a RetSat depletion should be provided : insulin resistance tests (ITT), glucose and pyruvate tolerance (GTT, PTT) tests should be performed. Insulin pulses with exogenous insulin should be performed in these mice to determine the precise insulin signaling cascade (IRS, Akt phosphorylation...) in liver.

2-Was the expression of RETSAT correlated with insulin resistance in humans? (HOMA-IR ?)

3-What is /are the mechanisms linking RetSat to ChREBP ? A direct effect ? An indirect effect through an effect on glucose metabolism? Glucose metabolites found to affect ChREBP expression/translocation should be measured. Overall the mechanism by which RetSat affects ChREBP expression/activity is not clear.

Reviewer #2 (Remarks to the Author):

The manuscript from Schupp and colleagues reports an interesting study of the actions of RetSat in liver disease. RetSat was first described about a decade ago and is known to promote adipogenesis, but a role for RetSat in liver disease has not been reported. Given the paucity of information regarding RetSat and its actions and the frequency at which bioinformatics studies have identified RetSat as the gene with the highest differential mRNA expression upon analysis of type 2 diabetes-related data sets, the present studies must be viewed as being highly significant. The authors' data indicate that RetSat depletion in primary hepatocytes reduces glycolytic flux, de novo lipogenesis and palmitate desaturation. The central conclusion the authors' draw from these data and other in vivo data obtained from mice and humans is that RetSat regulates hepatic metabolism through actions upstream of ChREBP.

The manuscript is well written and interesting. The authors' experimental approach is sound. However, the authors data are not always fully convincing. The data are generally believable but not all of the data are fully convincing of the authors' conclusions.

The authors conclude that RetSat depletion decreases de novo lipogenesis in primary mouse hepatocytes based on an experiment where labeled acetate incorporation into lipids was measured. Indeed, siRetSat treatment of the cells does lessen label incorporation into what is presumably a total lipid fraction extracted from hepatocytes, although this is not fully clear from the limited details provided by the authors. This experimental approach is one that is far from the gold standard used to assess de novo lipogenesis, one assessing the incorporation of labeled water into hepatic triglyceride pools. It will also be important to establish whether de novo lipogenesis is disrupted in livers of mice treated with siRetSat. These data are needed if the authors' claims regarding de novo lipogenesis are to be convincing.

There are concerns regarding the hepatic triglyceride levels reported by the authors. The text indicates that liver sections were first hydrolyzed for 5 h at 60°C in ethanolic 30% KOH and then TG determined. Would not this treatment result in saponification of the TG, possibly not all but some significant quantity? One wonders about the validity of these data. Were controls done to establish that this procedure does not give rise to TG hydrolysis?

Response to reviewers

Reviewer #1:

We would like to thank the reviewer for her/his helpful comments and suggestions to improve our manuscript. We performed additional data analyses and new experiments in order to address the concerns raised and hope that the revised manuscript is satisfactory.

1-Metabolic and physiologic analysis of mice with a RetSat depletion should be provided : insulin resistance tests (ITT), glucose and pyruvate tolerance (GTT, PTT) tests should be performed. Insulin pulses with exogenous insulin should be performed in these mice to determine the precise insulin signaling cascade (IRS, Akt phosphorylation...) in liver.

We thank the reviewer for pointing out these missing metabolic characterizations. We performed glucose and insulin tolerance tests in mice with liver-specific RetSat depletion. We found that glucose tolerance was indeed improved (**new Supplementary Fig. 2a**) whereas the insulin-induced reduction in blood glucose was similar (**new Supplementary Fig. 2b**). We also carried out the suggested stimulation of the insulin signaling pathway and found that Akt phosphorylation (P-Akt^{S473}) in liver after injecting insulin was comparable, and at basal levels slightly increased in shRetSat-treated mice (**new Supplementary Fig. 2c**). These results are now described on page 7 of the revised manuscript.

2- Was the expression of RETSAT correlated with insulin resistance in humans? (HOMA-IR ?)

The reviewer raises an interesting point. We found that RETSAT expression in human liver does correlate with the insulin resistance index HOMA-IR ($r^2 = 0.142$, $P = 0.044$, **new Fig. 3c**) and describe these findings on page 7 of the revised manuscript. A plausible explanation for this could be the identified relationship between RETSAT expression and liver lipid accumulation, as the latter is considered as one of the determining factors of insulin resistance (Shulman, 2014).

3-What is /are the mechanisms linking RetSat to ChREBP ? A direct effect ? An indirect effect through an effect on glucose metabolism? Glucose metabolites found to affect ChREBP expression/translocation should be measured. Overall the mechanism by which RetSat affects ChREBP expression/activity is not clear.

We thank the reviewer for valuable suggestions to further elucidate the underlying mechanisms. We quantified relevant glucose metabolites by mass spectrometry and found that most of the analyzed metabolites were increased in RetSat-depleted hepatocytes (**new Supplementary Fig. 6**). This is a valuable mechanistic insight and shows that reduced ChREBP activity upon RetSat depletion is not caused by the relative lack of activating metabolites like glucose 6-phosphate. Moreover, since ChREBP knockdown in liver (Dentin et al., 2006) or its whole-body deletion (Iizuka et al., 2006) is known to increase the abundance of glucose metabolites like glucose 6-phosphate in liver, it supports our findings in regard to the newly-discovered relationship between RetSat and ChREBP. Therefore, we hypothesize that RetSat depletion alters the abundance of other (non-sugar) metabolites known to regulate ChREBP activity (such as certain saturated/ unsaturated fatty acids (Dentin et al., 2005)) and/or affects ChREBP's post-translational modifications controlling its activity (Filhoulaud et al., 2013), as we discuss in the manuscript. A detailed dissection of these complex pathways and its regulation by RetSat will remain for future studies. A short summary of these aspects is included on page 10 of the revised manuscript.

References

- Dentin, R., Benhamed, F., Hainault, I., Fauveau, V., Fougelle, F., Dyck, J.R., Girard, J., and Postic, C. (2006). Liver-specific inhibition of ChREBP improves hepatic steatosis and insulin resistance in ob/ob mice. *Diabetes* 55, 2159-2170.
- Dentin, R., Benhamed, F., Pegorier, J.P., Fougelle, F., Viollet, B., Vaulont, S., Girard, J., and Postic, C. (2005). Polyunsaturated fatty acids suppress glycolytic and lipogenic genes through the inhibition of ChREBP nuclear protein translocation. *J Clin Invest* 115, 2843-2854.
- Filhoulaud, G., Guilmeau, S., Dentin, R., Girard, J., and Postic, C. (2013). Novel insights into ChREBP regulation and function. *Trends Endocrinol Metab* 24, 257-268.
- Iizuka, K., Miller, B., and Uyeda, K. (2006). Deficiency of carbohydrate-activated transcription factor ChREBP prevents obesity and improves plasma glucose control in leptin-deficient (ob/ob) mice. *Am J Physiol Endocrinol Metab* 291, E358-364.
- Shulman, G.I. (2014). Ectopic fat in insulin resistance, dyslipidemia, and cardiometabolic disease. *N Engl J Med* 371, 1131-1141.

Reviewer #2:

First of all, we would like to thank the reviewer for acknowledging the high relevance of our study and her/his helpful suggestions to improve the manuscript. We also thank the reviewer for commenting on a well written and interesting manuscript. The revised manuscript now includes experimental data that address the raised concerns.

The authors conclude that RetSat depletion decreases de novo lipogenesis in primary mouse hepatocytes based on an experiment where labeled acetate incorporation into lipids was measured. Indeed, siRetSat treatment of the cells does lessen label incorporation into what is presumably a total lipid fraction extracted from hepatocytes, although this is not fully clear from the limited details provided by the authors. This experimental approach is one that is far from the gold standard used to assess de novo lipogenesis, one assessing the incorporation of labeled water into hepatic triglyceride pools. It will also be important to establish whether de novo lipogenesis is disrupted in livers of mice treated with siRetSat. These data are needed if the authors' claims regarding de novo lipogenesis are to be convincing.

We thank the reviewer for these valuable comments on methodological issues regarding the analysis of *de novo* lipogenesis. We completely agree that the gold standard for analyzing lipogenesis is the incorporation of a deuterated water label into the liver triglyceride pool. We performed this experiment and found that palmitate synthesis in mice with hepatic RetSat depletion was reduced (**new Fig. 3f**, $P = 0.056$ vs. sh β Gal control mice). This is supportive of our other data that implicate RetSat in the regulation of hepatic lipogenesis.

In regard to *in vitro* assays that determine lipogenesis, we believe that measuring incorporation of the radioactively labeled substrate ^{14}C -acetate into extractable (total) lipids is an applicable readout for lipogenic flux, especially in rather small samples of cellular material like the here investigated cultured primary hepatocytes. As commented on by the reviewer, we now provide more details on the description of the method, which can be found on page 17 of the revised manuscript. Notably, a number of other studies have taken advantage of this assay (Akie and Cooper, 2015; Bedu et al., 2002; Guillet-Deniau et al., 2004; Hagiwara et al., 2012; Harada et al., 2007; Stefanovic-Racic et al., 2008; Wendel et al., 2013), and we ourselves have validated this method by comparing lipogenesis in hepatocytes with or without insulin or in cultured preadipocytes/adipocytes with the expected outcomes.

There are concerns regarding the hepatic triglyceride levels reported by the authors.

The text indicates that liver sections were first hydrolyzed for 5 h at 60°C in ethanolic 30% KOH and then TG determined. Would not this treatment result in saponification of the TG, possibly not all but some significant quantity? One wonders about the validity of these data. Were controls done to establish that this procedure does not give rise to TG hydrolysis?

We apologize for this confusing method description. Yes, treatment of liver samples with ethanol/KOH for 5 h at 60 °C releases and hydrolyzes tissue triglycerides. The subsequent measurement is indeed performed with a 'triglyceride kit' that takes advantage of the commonly used glycerol 3-phosphate oxidase reaction (the substrate being generated from hydrolyzed triglycerides) with a coupled colorimetric detection. Since the amount of released glycerol is proportional to the initial content of tissue triglycerides, this method allows their indirect quantification. We have clarified the method description accordingly on page 15 of the revised manuscript.

References

- Akie, T.E., and Cooper, M.P. (2015). Determination of Fatty Acid Oxidation and Lipogenesis in Mouse Primary Hepatocytes. *J Vis Exp*, e52982.
- Bedu, E., Chainier, F., Sibille, B., Meister, R., Dallevet, G., Garin, D., and Duchamp, C. (2002). Increased lipogenesis in isolated hepatocytes from cold-acclimated ducklings. *Am J Physiol Regul Integr Comp Physiol* 283, R1245-1253.
- Guillet-Deniau, I., Pichard, A.L., Kone, A., Esnous, C., Nieruchalski, M., Girard, J., and Prip-Buus, C. (2004). Glucose induces de novo lipogenesis in rat muscle satellite cells through a sterol-regulatory-element-binding-protein-1c-dependent pathway. *J Cell Sci* 117, 1937-1944.
- Hagiwara, A., Cornu, M., Cybulski, N., Polak, P., Betz, C., Trapani, F., Terracciano, L., Heim, M.H., Ruegg, M.A., and Hall, M.N. (2012). Hepatic mTORC2 activates glycolysis and lipogenesis through Akt, glucokinase, and SREBP1c. *Cell Metab* 15, 725-738.
- Harada, N., Oda, Z., Hara, Y., Fujinami, K., Okawa, M., Ohbuchi, K., Yonemoto, M., Ikeda, Y., Ohwaki, K., Aragane, K., Tamai, Y., and Kusunoki, J. (2007). Hepatic de novo lipogenesis is present in liver-specific ACC1-deficient mice. *Mol Cell Biol* 27, 1881-1888.
- Stefanovic-Racic, M., Perdomo, G., Mantell, B.S., Sipula, I.J., Brown, N.F., and O'Doherty, R.M. (2008). A moderate increase in carnitine palmitoyltransferase 1a activity is sufficient to substantially reduce hepatic triglyceride levels. *Am J Physiol Endocrinol Metab* 294, E969-977.
- Wendel, A.A., Cooper, D.E., Ilkayeva, O.R., Muoio, D.M., and Coleman, R.A. (2013). Glycerol-3-phosphate acyltransferase (GPAT)-1, but not GPAT4, incorporates newly synthesized fatty acids into triacylglycerol and diminishes fatty acid oxidation. *J Biol Chem* 288, 27299-27306.

REVIEWERS' COMMENTS:

Reviewer #1 (Remarks to the Author):

No more comments

Reviewer #2 (Remarks to the Author):

The authors have addressed well my earlier comments. I have nothing new to convey.